# OpenReview forum: "Teacher's pet: understanding and mitigating biases in distillation"
_ICLR.cc/2022/Conference — ICLR 2022 Submitted_

### Official Review · Reviewer_bvA3 · 2021-10-22

**Correctness:** 3
**Technical Novelty And Significance:** 3
**Empirical Novelty And Significance:** 3
**Recommendation:** 6
**Confidence:** 4

**Main Review:**


+ The paper is well-written and easy to follow and brings out an interesting issue with distillation
+ An easy to implement solution is proposed for addressing the issue highlighted.
- It is not clear if the same classes are "problematic" for each model. In other works, if I train a model 10 times do we expect the drop in performance to be in the same 10 classes?
- I would expect a more detailed statistical analysis to validate that there is indeed a consistent decrease in classes in a statistically significant manner. This should include training multiple models and evaluating their performance.
- When measuring the worst1/10 performance, we measure the performance on the initial worst 10 classes or there might be a reduction in the performance in other classes? In any case, the average performance is increasing. However, it would be useful to know if the proposed method can negatively impact some classes.
- AdaAlpha/AdaMargin are not compared with more advanced distillation methods. I understand that the issue highlighted by the authors is important. However, it is not clear if this also affects the more recent distillation approaches (e.g., those performing online distillation, distillation from ensembles, using knowledge from intermediate layers, etc.) as well or if it only concerns vanilla distillation.


**Summary Of The Paper:**

In this work the authors examine the impact of distillation on certain sub-groups of the data. More specifically, they demonstrate that even though distillation can increase the overall accuracy of a model, it can harm the accuracy on certain subclasses. This includes both self-distillation, as well as regular distillation setups. To overcome this, the authors propose using class-specific weights, that have been tuned according to a holdout set, to tune the importance of each class during distillation, as well as using softmax cross-entropy with margins tuned according to the difficulty of each class. Furthermore, it is also demonstrated that these phenomena can also impact the fairness of the model.


**Summary Of The Review:**

Overall, I enjoyed reading this paper, which brings out an important issue in distillation. However, I would feel more confident in the reported results if a more detailed statistical analysis was provided, as well as some comparisons with more recent distillation approaches, which leaves me with many open questions. Furthermore, there is no discussion on why distillation brings out this behavior from a theoretical standpoint.

---

> ### Author Response · Authors · 2021-11-17
> **Response to Reviewer bvA3**
>
> Thanks for the detailed comments. We are glad the reviewer appreciated the work.
>
> > It is not clear if the same classes are "problematic" for each model. In other works, if I train a model 10 times do we expect the drop in performance to be in the same 10 classes?
>
> We did observe the drop under distillation to be consistently over the hardest classes, with a slight variation across classes. We plot an example illustration of this phenomenon in Figure 6 in the appendix.
>
> > I would expect a more detailed statistical analysis to validate that there is indeed a consistent decrease in classes in a statistically significant manner. This should include training multiple models and evaluating their performance.
>
> We did run each reported distillation experiment 3 times, but omitted standard deviations for readability. We did observe the statistical significance of the drops in worst class performance upon distillation. For example, for CIFAR-100 LT, we get: one-hot avg acc: 43.22+/-0.05, worst-10 acc: 2.33+/-0.40; distilled avg acc: 45.39+/-0.14, worst-10 acc: 0.87+/-0.12.
>
> > When measuring the worst1/10 performance, we measure the performance on the initial worst 10 classes or there might be a reduction in the performance in other classes? In any case, the average performance is increasing. However, it would be useful to know if the proposed method can negatively impact some classes.
>
> Indeed, we did notice distillation to significantly improve the easiest classes compared to no distillation baseline. As the reviewer rightly pointed out, despite the drop in worst class performance, the average increases, and so there are certain classes where distillation helps. Our methods often strike a trade off where worst class performance is improved at the expense of easier classes. We added a clarification about this to Section 3.1.
>
> > AdaAlpha/AdaMargin are not compared with more advanced distillation methods. I understand that the issue highlighted by the authors is important. However, it is not clear if this also affects the more recent distillation approaches (e.g., those performing online distillation, distillation from ensembles, using knowledge from intermediate layers, etc.) as well or if it only concerns vanilla distillation.
>
> We agree a study of the approaches mentioned by the reviewer would be very interesting. We would like to point out that logit based distillation we consider is the current state of the art in distillation, as achieved by Beyer et al. 2021 https://arxiv.org/pdf/2106.05237.pdf (82.8 accuracy for ResNet-50).

---

> > ### Comment · Reviewer_bvA3 · 2021-11-18
> > **Reviewer Response**
> >
> > I would like to thank the authors for their response. Authors addressed some of my concerns, which can improve the quality of the paper. I think it is important to add the additional experimental data in the Appendix, if the paper is accepted.

---

### Official Review · Reviewer_t1Uw · 2021-10-29

**Correctness:** 4
**Technical Novelty And Significance:** 2
**Empirical Novelty And Significance:** 2
**Recommendation:** 5
**Confidence:** 5

**Main Review:**

The paper is a pleasure to read, very well written, clear and convincing with their arguments, and do some good work (with small caveats) regarding the experimental support of the claims. I also like that ImageNet is thoroughly used rather than sticking to datasets like CIFAR.

My main issues however are 1) while biases in learning is clearly a topic that has relevance nowadays, the focus of this paper ends up being on some specific aspect of a specific technique that might cause bias amplification. Thus, the focus is quite narrow 2) The results are not impressive. Performance improvements, even on the specific focus group (lower-performing classes) the gains are not very strong or consistent. In more detail:

Regarding 1, generally speaking, I think the community has too much of a fascination with the logit matching knowledge distillation method. Knowledge distillation is widely used and very important, but the specific logit matching flavour of KD is a 2015 paper that has been thoroughly improved multiple times. I am always a bit confused about why the community keeps producing a large corpus of works dissecting small aspects of logit matching KD, while forgetting about the whole literature that followed that seminal paper. (I have to admit here though that logit matching KD still performs admirably against some of the other methods that are supposed to work better, and that it also tends to give gains when combining with other KD methods).

Still, nowadays the state of the art is well above logit matching KD. I would be much more interested in the paper if it was applicable across KD methods (or at least on top of a couple of state of the art methods). For example, is it relevant for feature distillation (Heo et al., A Comprehensive Overhaul of Feature Distillation, ICCV'19)? or can it be combined with Xu et al., Knowledge Distillation Meets Self-Supervision, ECCV'20?

Issue 2: The main results, shown in Table 3, are not very impressive. On ImageNet, (logit matching) distillation marginally hurts overall performance. The proposed method shows that it can keep the performance to roughly "no distillation" level, and for ImageNet LT the worst-1 and worst-10 accuracy are basically 0% for all (thus no improvement) and the worst-100 actually takes a small hit (quite irrelevant given the overall poor performance, but still). Also, the average performance is improved marginally. So all in all, the CIFAR-100 are the only convincing set of results.

Comparison with other approaches is also somewhat absent, e.g. Zhou et al (2021) (I think the most related method?) is only compared in one setting, and a table with a wide set of comparisons against other state of the art distillation methods is also absent (see for example Tables 3 and 4 of the aforementioned Zhou et al (2021) paper).


Other aspects (not really important for the overall score and not necessary to reply):
- On Fig. 1 caption: shouldn't it be 2% absolute instead of 1% absolute?
- Page 2, bottom: the effect of the temperature is different (inverted) if > 1 or < 0, so the last phrase of the page is incorrect.
- Eq. 2 is quite trivial, I'm not sure it is necessary to spell it out.
- Section 3.1, the hyperparameter choices are not in appendix C, but in B.2 I believe.
- Same paragraph, +0.4 improvement over the baseline is hardly a success... I'm not sure this kind of claim is too healthy
- Table 1 could do with a CIFAR baseline so we can compare with line 3. Also, when comparing the long tail behavior with the standard behavior, we see less variance (less avg. gain, better worse performance), which feels kind of contradictory with the train of thought of the paper?
- The last paragraph of section 3.4 looks unsubstantiated to me.
- The analysis, especially for the experiments involving less classes, could do with some statistics - if the per-class performance is drawn from a normal distribution, it is clear that the worst performing one will depend on the number of classes (i.e. samples drawn from the distribution), so it is not surprising that reducing the number of classes of ImageNet means the performance of the worse class improves.
- It feels like the formulation is calling for a per-sample weighting... in the end the average per-class performance is used as proxy but it seems to me that what matters is the behavior of the teacher for a specific instance.
- Zhou et al. (2021) looks like the most related work, but I feel it is not explained properly in the related work. Explaining that method better, and highlighting the differences to the current proposal would improve the paper in my opinion.



**Summary Of The Paper:**

The paper proposes to have a closer look at (logit matching) knowledge distillation, and explores whether the gains are uniform across all portions of the test data or if instead there are gains on some portions of the data but damaged performance on others. This is justified by the potential of KD to amplify biases. The authors quickly show that indeed the gains are not uniform, some classes being affected negatively by KD so that the bad performance of the teacher on some classes is amplified on the student. Then the authors conduct a number of experiments to study the impact of different aspects of training (e.g. architectures involved and aspects of data like class imbalance) over this amplification of class-specific poor performance. They are then able to identify what classes are likely to be negatively affected and follow that with the proposal of two modifications of the basic KD formulation with the aim to improve KD effect on the worse-performing classes. Experiments focus on CIFAR-100 and ImageNet, including long-tail variants.

**Summary Of The Review:**

The paper has some interesting aspects and does a good deep-dive into some very specific aspect of knowledge distillation, but the focus is very narrow and the improvements obtained do not look impressive enough. The score would be around 4 if that existed.

---

> ### Author Response · Authors · 2021-11-17
> **Response to Reviewer t1Uw**
>
> Thanks for the detailed comments. We are glad the reviewer finds that "the paper is a pleasure to read".
>
> > while biases in learning is clearly a topic that has relevance nowadays, the focus of this paper ends up being on some specific aspect of a specific technique that might cause bias amplification. Thus, the focus is quite narrow.
>
> We would like to point out that logit based distillation is the current state of the art in distillation, as achieved by Beyer et al. 2021 https://arxiv.org/pdf/2106.05237.pdf (82.8 accuracy for ResNet-50).
>
> Moreover, Zhou et al. 2021 who employ per example weighting in the logit-based matching distillation, report outperforming  (Heo et al., A Comprehensive Overhaul of Feature Distillation, ICCV'19), mentioned by the reviewer.
>
> > The results are not impressive. Performance improvements, even on the specific focus group (lower-performing classes) the gains are not very strong or consistent.
>
> We indeed do not see gains across all considered datasets. We emphasize that a bulk of the paper comprises an analysis of *when* and *why* distillation harms subgroups. We view this as our primary contribution. The bulk of the paper (pages 3–6) is on analysis of the *causes* of non-uniform gains in distillation in Section 3, whereas we only spend a single page discussing new methods in Section 4.
>
> > It is clear that the worst performing one will depend on the number of classes (i.e. samples drawn from the distribution), so it is not surprising that reducing the number of classes of ImageNet means the performance of the worse class improves.
>
> The reviewer’s comment makes sense when considering absolute per class performance from a model. However, we are not investigating absolute per-class accuracies, but rather *differences* in per class accuracies between distilled and one hot student models. We observe how the hardest classes suffer from a disproportionately worsened accuracy from the distilled model.

---

> > ### Comment · Reviewer_t1Uw · 2021-11-29
> > **Reply to author's rebuttal**
> >
> >
> > Regarding the KD method, Zhou et al. have very solid results, but also change the objective from the original logit-matching KD paper the current submission builds upon. So they are related, but that does not mean that results translate. I have to admit I wasn't aware of the Beyer et al. paper, which is an arxiv paper afaik. However, after reading it, there is little insight to be had: they get a huge teacher, train for 10k epochs on ImageNet and use mixup. I think it has some level of valuable insight due to the huge number of experiments, but it is just the furthest from an apples to apples comparison to other KD methods there is.
> >
> > In any case, I see little extra information in the rebuttal to change the recommendation. All in all, the paper has some pluses, but due to the reasons highlighted in my original review I still don't think it deserves being accepted.

---

> > > ### Author Response · Authors · 2021-11-29
> > > **Response to Reviewer t1Uw**
> > >
> > > We thank the reviewer for the response.
> > >
> > > >Zhou et al. have very solid results, but also change the objective from the original logit-matching KD paper the current submission builds upon. So they are related, but that does not mean that results translate.
> > >
> > > If the reviewer considers Zhou et al. to be a different method than logit matching, we would like to highlight we did provide results in the paper on Zhou et al. and show how it also suffers from disproportionately worsened accuracy. In particular, worst-1 accuracy drops by -0.56 compared to the one-hot student, see Table 4.

---

> > > > ### Comment · Reviewer_t1Uw · 2021-11-29
> > > > **clarification**
> > > >
> > > >
> > > > I got a bit sidetracked with the discussion about the different methods... I am usually a tad surprised that a lot of effort is put into insights focusing on the original KD formulation in Hinton et al. when there's many other KD variants that receive none of that attention. We can call Zhou et al. a different method or a different flavour, it does not change my point. For example, the main results on the current submission are on Hinton et al. algorithm instead of Zhou et al. But again, this is a bit of a side argument.
> > > >
> > > > Regarding the core issue (that you indeed have some results on Zhou et al), I have to admit I didn't understand this table fully when first reviewing the paper (I guess a miss on my original review). Zhou et al. has a weighting per example, but is the result on Table 4 Zhou et al. + yours or just Zhou et al as an alternative strategy to yours? Or a variant in which you apply your strategy in a per-sample manner instead of a per-class manner as in Zhou et al.?
> > > > To be honest I interpreted it as a per-instance application of your strategy, but I see how this might have been unsupported. Let's assume it is Zhou et al. + yours, it is still a bit unconvincing: for example we don't know whether the average performance is degraded (it seems like the Zhou et al. entry is low?).
> > > >
> > > > I was anyway trying to make a more general point: if you presented results that apply to any/a few KD methods it would have more general interest than when presenting it on one specific version of KD. I don't think Table 4 goes a long way in showing the generality of the method.

---

> > > > > ### Author Response · Authors · 2021-11-29
> > > > > **Response to Reviewer t1Uw**
> > > > >
> > > > > We agree that studying multiple KD methods would be interesting. However, space is finite, and even for a single KD method, there are many things to study: does it harm under cross-architecture, small # of classes, label imbalance, ... Adding another KD method means adding yet another dimension to the study. We think the study is as it is pretty packed at the moment.
> > > > >
> > > > > Our point regarding: Beyer et al. and Zhou et al. is simply that matching logits, as opposed to intermediate features, is not fundamentally limited. We again agree that studying other methods would be interesting, but see it as fruitful for future work.

---

### Official Review · Reviewer_pMdP · 2021-10-30

**Correctness:** 4
**Technical Novelty And Significance:** 4
**Empirical Novelty And Significance:** 4
**Recommendation:** 6
**Confidence:** 5

**Main Review:**

**Strengths**

The identified problem, distillation performance on different subgroups, is highly related to the fairness of machine learning, which is an interesting and important research topic. Experiments on balanced and long-tailed datasets further evaluate how distillation performs diversely on different subgroups.

This paper proposed two methods for improving subgroup performance, AdaWeight with adaptive mixing weights and AdaMargin with per-class margins.

This paper is clearly written and well organized. The writing quality is satisfying.

**Weaknesses**

The improvements of AdaAlpha and AdaMargin on ImageNet LT, a larger dataset for long-tailed classification, are marginal. Specifically, AdaAlpha achieves only an improvement of 0.15 mean per-class accuracy, and the training of AdaMargin diverges. It is not clear whether AdaAlpha works on large datasets. This marginal improvement limits the contribution of the proposed methods. Another large-scale dataset, iNaturalist 2018, may be helpful to further evaluate the effectiveness of AdaAlpha and AdaMargin.

The contribution of AdaMargin is somehow limited. As shown in Table 3, AdaMargin achieves a significant improvement of worst-1/10 per-class accuracy on CIFAR-100. However, the improvement on ImageNet compared to one-hot and distillation are very marginal. Also, the training of ImageNet LT diverges. These results show that AdaMargin is not a good choice in practice, which limits its technical contribution.

For the long-tailed setting, this paper evaluated the performance on worst-1, worst-10, worst-100 subgroups. The popular subgroups in long-tailed recognition are many-shot, medium-shot and few-shot. It is okay to choose not to report the performances on the normal subgroups, but an explanation on why not using these subgroups is welcomed.

**Summary Of The Paper:**

This paper identified an underexplored problem with distillation, fairness of distillation on different subgroups. This paper further proposed two methods to tackle this challenge. Experiments are conducted on CIFAR-100, ImageNet, and their long-tailed variants.

**Summary Of The Review:**

Overall, this paper proposed an interesting and important perspective on the fairness of distillation, which will inspire future works to pay more attention to the fairness of distillation. The writing quality is good. The contributions to empirical analysis are acknowledged by my side.

The weakness comes from the technical contributions. In particular, it seems that the proposed AdaMargin does not work well in practice. This limits the contributions of the methods part. The contribution of AdaAlpha is clear.

Considering both the strengths and weaknesses, I vote for acceptance at this stage. My initial score is between 6 and 8.

---

> ### Author Response · Authors · 2021-11-17
> **Response to Reviewer pMdP**
>
> Thanks for the detailed comments. We are glad the reviewer appreciated the work.
>
> > The improvements of AdaAlpha and AdaMargin on ImageNet LT, a larger dataset for long-tailed classification, are marginal.
>
> We indeed do not see gains across all considered datasets. We emphasize that a bulk of the paper comprises an analysis of *when* and *why* distillation harms subgroups. We view this as our primary contribution. The bulk of the paper (pages 3–6) is on analysis of the *causes* of non-uniform gains in distillation in Section 3, whereas we only spend a single page discussing new methods in Section 4.
>
> > The contribution of AdaMargin is somehow limited.
>
> We agree that AdaAlpha showed more promising results. As noted in section 4.2, AdaMargin is inspired by the long tailed literature, and so it seemed like a natural approach to consider.
>
> > For the long-tailed setting, this paper evaluated the performance on worst-1, worst-10, worst-100 subgroups. The popular subgroups in long-tailed recognition are many-shot, medium-shot and few-shot. It is okay to choose not to report the performances on the normal subgroups, but an explanation on why not using these subgroups is welcomed.
>
> We considered worst-k accuracies for consistency across datasets, both long tail and non-long tail.

---

> > ### Comment · Reviewer_pMdP · 2021-11-26
> > **Re: Response to Reviewer pMdP**
> >
> > Thank the authors for their response.
> >
> > After reading the authors' response and other reviewers' comments, I decided to finalize my score as 6 and regard this paper as borderline.
> >
> > I acknowledged the contribution of this work. Especially, Reviewer QhBi pointed out that the topic is already known in distillation and the paper is like an experimental report, and the authors provided feedback to this concern. I am on the authors' side and look forward to further comments from Reviewer QhBi.
> >
> > However, my concerns that (1) the improvements are marginal and (2) the contribution of AdaMargin is limited, are not well addressed. These are the two main limitations of this paper.
> >
> > Considering both the strengths and weaknesses, I will rate for 6, a borderline accept. I would not fight for acceptance if the paper is rejected.

---

### Official Review · Reviewer_QhBi · 2021-11-03

**Correctness:** 3
**Technical Novelty And Significance:** 2
**Empirical Novelty And Significance:** 2
**Recommendation:** 3
**Confidence:** 5

**Main Review:**

- The authors aim at improving the performance of the underperforming groups of classes, which can be easily observed. Thus, for me, this observation is not surprising enough. So in section 3 (are distillation’s gains uniform), the authors use big space of the main paper to tell us the conclusions, which are already known by the researchers in the field of distillation. These conclusions are mainly roughly drawn from the simple experiments and lack the theoretical support. In this way, this paper is more or less like an experimental report.

- In the main paper, it is said that the labels that are not well learned by the teacher performance even worse in the student network with direct distillation. But I just wonder if it is also a kind of knowledge to be learned? Is it necessary to just simply eliminate this?

- The proposed two methods are too straightforward and intuitive and not new. So, the contribution of the main method is limited.


- Then about the experiments. Does the author mainly focus on the self-distillation? I don’t think in the situation of self-distillation, the problem of the unbalanced performance should be well focused. What about the experiments in transfer learning? There are more poorly performed labels.




**Summary Of The Paper:**

- This paper is established by a discovered phenomenon that in distillation, not all the classes’ performance is improved although the student is improved significantly.

- Thus, the authors explore a method that focuses on promoting the performance of the poorly-performed labels, which also lead to the overall promotion while distillation.

- The authors propose two methods to achieve it, one is the distillation with adaptive mixing weights and the other is the distillation with per-class margins.


**Summary Of The Review:**

- please restate the contribution of the proposed method.

- should need more experiments. like in transfer learning. And need more analysis, like what is the performance of the best-performance group of labels.

---

> ### Author Response · Authors · 2021-11-17
> **Response to Reviewer QhBi**
>
> Thanks for the detailed comments.
>
> > which are already known by the researchers in the field of distillation
>
> We are not aware of _any_ prior systematic analysis of the potential non-uniform gains of distillation, nor the causes of the same. We would appreciate any references for the reviewer’s claim.
>
> > These conclusions are mainly roughly drawn from the simple experiments and lack the theoretical support. In this way, this paper is more or less like an experimental report.
>
> We agree that the paper’s analysis is empirical rather than theoretical. However, we emphasise that:
>
> 1) Our focus is on systematic empirical _analysis_ with the goal of understanding a non-trivial phenomenon, rather than merely empirical _comparison_: Sec 3 is devoted to carefully understanding the extent of, and causes for, the non-uniform gains of distillation.
>
> 2) Works that empirically analyse neural network phenomena have been well-received in the core ML community, e.g.,
> Zhang et al., “Understanding deep learning requires rethinking generalization”, ICLR 2017;
> Muller et al., “When Does Label Smoothing Help?”, NeurIPS 2019;
> Nakkiran et al., “Deep Double Descent: Where Bigger Models and More Data Hurt”, ICLR 2019;
> Neyshabur et al., “What is being transferred in transfer learning?”, NeurIPS 2020;
> Jiang et al., “Characterizing Structural Regularities of Labeled Data in Overparameterized Models”, ICML 2021.
>
> We added a clarification about this in the Introduction section.
>
> > In the main paper, it is said that the labels that are not well learned by the teacher performance even worse in the student network with direct distillation. But I just wonder if it is also a kind of knowledge to be learned? Is it necessary to just simply eliminate this?
>
> It is true that the teacher's uncertainty about labels may constitute potentially useful information. However, this information is not successfully used in distillation experiments, as the distilled student’s classification error on hard classes increases. Mitigation of this effect is useful if one cares about performing well across classes.
>
> > The proposed two methods are too straightforward and intuitive and not new. So, the contribution of the main method is limited.
>
> We emphasize that a bulk of the paper comprises an analysis of *when* and *why* distillation harms subgroups. We view this as our primary contribution. The bulk of the paper (pages 3–6) is on analysis of the *causes* of non-uniform gains in distillation in Section 3, whereas we only spend a single page discussing new methods in Section 4.
>
> > Then about the experiments. Does the author mainly focus on the self-distillation? I don’t think in the situation of self-distillation, the problem of the unbalanced performance should be well focused. What about the experiments in transfer learning? There are more poorly performed labels.
>
> We focused on the teacher - student distillation where data for both is the same set of examples. This is the setup widely considered in the core ML community when studying fundamental properties of distillation, e.g., Menon et al. 2020, Zhou et al. 2021, Beyer et al. 2021. Please note we not only consider self-distillation, which is commonly referred to as a setup where teacher and student architectures are the same, but instead consider a wide range of teacher and student architectures (e.g. see Table 2).

---

### Decision · Program_Chairs · 2022-01-20

**Decision:**

Reject

**Comment:**

This paper studies knowledge distillation and explores why distillation gains are not uniform. Reviewers consistently find this paper an interesting read, but had common concerns on generalizability and limited improvements/contributions.
In general, reviewers mostly gave a score that is below the acceptance threshold, or expressed concerns otherwise. Summing these up, we conclude this paper is of interest to the ICLR audience, but current form is not ready yet for acceptance.

Summary Of Reasons To Publish:
interesting analysis of the causes of non-uniform gains in distillation

Summary Of Suggested Revisions:

 (1) the improvements are marginal and (2) the contribution of AdaMargin is limited, (3) generalizability to other KDs